# Rhythmic circuit function is more robust to changes in synaptic than intrinsic conductances

Zachary Fournier[†], Leandro M Alonso*, Eve Marder*

Volen Center and Biology Department, Brandeis University, Waltham, United States

## eLife Assessment

This **important** study provides **compelling** insights into the differential impact of intrinsic and synaptic conductances on circuit robustness using computational models of the pyloric network from the crustacean stomatogastric ganglion. The results demonstrate that model networks are more sensitive to perturbations in intrinsic conductances than in synaptic conductances, highlighting the critical role of intrinsic plasticity in stabilizing neuronal networks. These findings underscore the importance of intrinsic plasticity, a crucial yet often overlooked factor in neuronal dynamics. The generality of these conclusions should be tested across diverse networks and functions.

*For correspondence:
lalonso@brandeis.edu (LMA);
marder@brandeis.edu (EM)

Present address: [†]Department of Biochemistry and Molecular Biology, The University of Chicago, Chicago, United States

Competing interest: The authors declare that no competing interests exist.

**Abstract** Circuit function results from both intrinsic conductances of network neurons and the synaptic conductances that connect them. In models of neural circuits, different combinations of maximal conductances can give rise to similar activity. We compared the robustness of a neural circuit to changes in their intrinsic versus synaptic conductances. To address this, we performed a sensitivity analysis on a population of conductance-based models of the pyloric network from the crustacean stomatogastric ganglion (STG). The model network consists of three neurons with nine currents: a sodium current (Na), three potassium currents (Kd, KCa, KA), two calcium currents (CaS and CaT), a hyperpolarization-activated current (H), a non-voltage-gated leak current (leak), and a neuromodulatory current (MI). The model cells are connected by seven synapses of two types, glutamatergic and cholinergic. We produced one hundred models of the pyloric network that displayed similar activities with values of maximal conductances distributed over wide ranges. We evaluated the robustness of each model to changes in their maximal conductances. We found that individual models have different sensitivities to changes in their maximal conductances, both in their intrinsic and synaptic conductances. As expected, the models become less robust as the extent of the changes increases. Despite quantitative differences in their robustness, we found that in all cases, the model networks are more sensitive to the perturbation of their intrinsic conductances than their synaptic conductances.

## Introduction

Since the early studies of *Hebb, 1949*; *Hopfield, 1982*; *Sejnowski and Tesauro, 1989*, it has been often assumed that changes in synaptic strength provide the salient mechanism by which circuit dynamics are altered in learning *Siegelbaum and Kandel, 1991*; *Dayan, 2001*; *Tsodyks et al., 1998*; *Abbott and Nelson, 2000*; *Sjöström et al., 2001*. This tradition was reinforced by years of extracellular recordings of spikes in brain circuits and by a myriad of computational studies using simplified model neurons such as rate models or integrate and fire neurons *Kempter et al., 1999*; *Fusi, 2002*; *Caporale and Dan, 2008*; *Fiebig and Lansner, 2017*. While all these studies have been extremely

informative, they have led to an underestimate of the role of intrinsic membrane currents in network dynamics in larger networks.

At the same time, it is now patently clear that most, if not all, brain circuits display many voltage and time-dependent currents that give their neurons interesting and complex cellular dynamics *McCormick and Huguenard, 1992*; *Markram, 1997*; *Ranjan et al., 2020*; *Grienberger and Magee, 2022*. Despite this, there are still prevailing beliefs that changes in synaptic strength that occur as a consequence of activity, represent the dominant mechanism for altering circuit dynamics. In contrast, those who have been studying the role of synaptic and intrinsic currents in circuit dynamics in small central pattern generating circuits have been acutely aware of the importance of modulating intrinsic currents *Calabrese and De Schutter, 1992*; *Swensen and Marder, 2000*; *Marder and Bucher, 2001*; *Harris-Warrick, 2011*; *Yang et al., 2022* and in some cases the relative insignificance of modifications of synaptic strength *Prinz et al., 2003a*. Nonetheless, to the best of our knowledge, there have not been systematic sensitivity analyses in circuits with easily measurable outputs to compare the relative importance of modification of synaptic and intrinsic conductances for network performance.

In this study, we take advantage of the stereotyped motor patterns of the crustacean pyloric rhythm *Maynard, 1972*; *Selverston, 1976* to assess its stability to changes in its conductances. Because we did not want these results to be an idiosyncratic consequence of the set of starting parameters, we generated 100 degenerate models (different sets of parameters with similar outputs) to explore the generality of the results in this basic circuit, and then evaluated the robustness of each of these models to changes in circuit parameters.

## Results

### Feature extraction to classify model network activity

The pyloric rhythm in crustaceans is produced by the sequential bursting of the PD (pyloric dilator), LP (lateral pyloric), and PY (pyloric) neurons. The AB (anterior burster) neuron is electrically coupled to the PD neurons and together they form a pacemaker kernel. In these simulations, the AB and PD neurons are represented by a single compartment cell, which we call PD. We modified the model of the pyloric network in *Prinz et al., 2004* to include a neuromodulatory current, $I_{MI}$, in each cell *Golowasch and Marder, 1992b*; *Swensen and Marder, 2000*. *Figure 1A* shows a schematic of the network studied in this work. Each cell is modeled as a single compartment with multiple currents *Golowasch and Marder, 1992a*; *Buchholtz et al., 1992*; *Goldman et al., 2001*. The interactions in the network are mediated by seven chemical synapses of two types: glutamatergic and cholinergic *Prinz et al., 2004*. Each neuron has a sodium current, $I_{Na}$; transient and slow calcium currents, $I_{CaT}$ and $I_{CaS}$; a transient potassium current, $I_A$; a calcium-dependent potassium current, $I_{KCa}$; a delayed rectifier potassium current, $I_{Kd}$; a hyperpolarization-activated inward current, $I_H$; a leak current, $I_{leak}$; and a neuromodulatory current $I_{MI}$.

The voltage traces in *Figure 1B* are an example of normal triphasic pyloric activity produced by one of the models in this study. To evaluate the activity of the network, we performed a number of measurements on the voltage traces of each cell. The colored lines correspond to the quantities used to calculate features of the network's activity. Duty cycles are calculated as the burst duration $B_d$ divided by the bursting period, $\tau_b$, of the cell. The phases of the rhythm at which the cells begin and end their bursts, are measured using the start of the PD burst as a reference (*Figure 1B*). Because the network activity is approximately periodic, it is possible to normalize the times of these events by the period of the networks activity $\tau_n = \tau_b$. In this way, we computed a number between 0 and 1 that corresponds to the ON and OFF phases for each cell. By computing these quantities for several periods of the activity, we obtained a distribution of values of duty cycles, burst periods, and ON/OFF phases, for each cell. The mean values and standard deviations of these distributions were used as *features* for classification purposes. For example, a cell that produces irregular bursts will yield a distribution of duty cycles with a standard deviation that would be larger than in the case of a cell that bursts regularly in a normal triphasic rhythm.

The activity of the network depends on the values of the maximal conductances $\bar{g}_i$, where $i$ is an index corresponding to the different current types (Na, CaS, CaT, Kd, KCa, KA, H, Leak, $I_{MI}$). In this study, we compare the sensitivity of the network to changes in intrinsic versus synaptic conductances. *Figure 1C* shows a normal (control) triphasic rhythm in black. The blue trace in *Figure 1C* corresponds

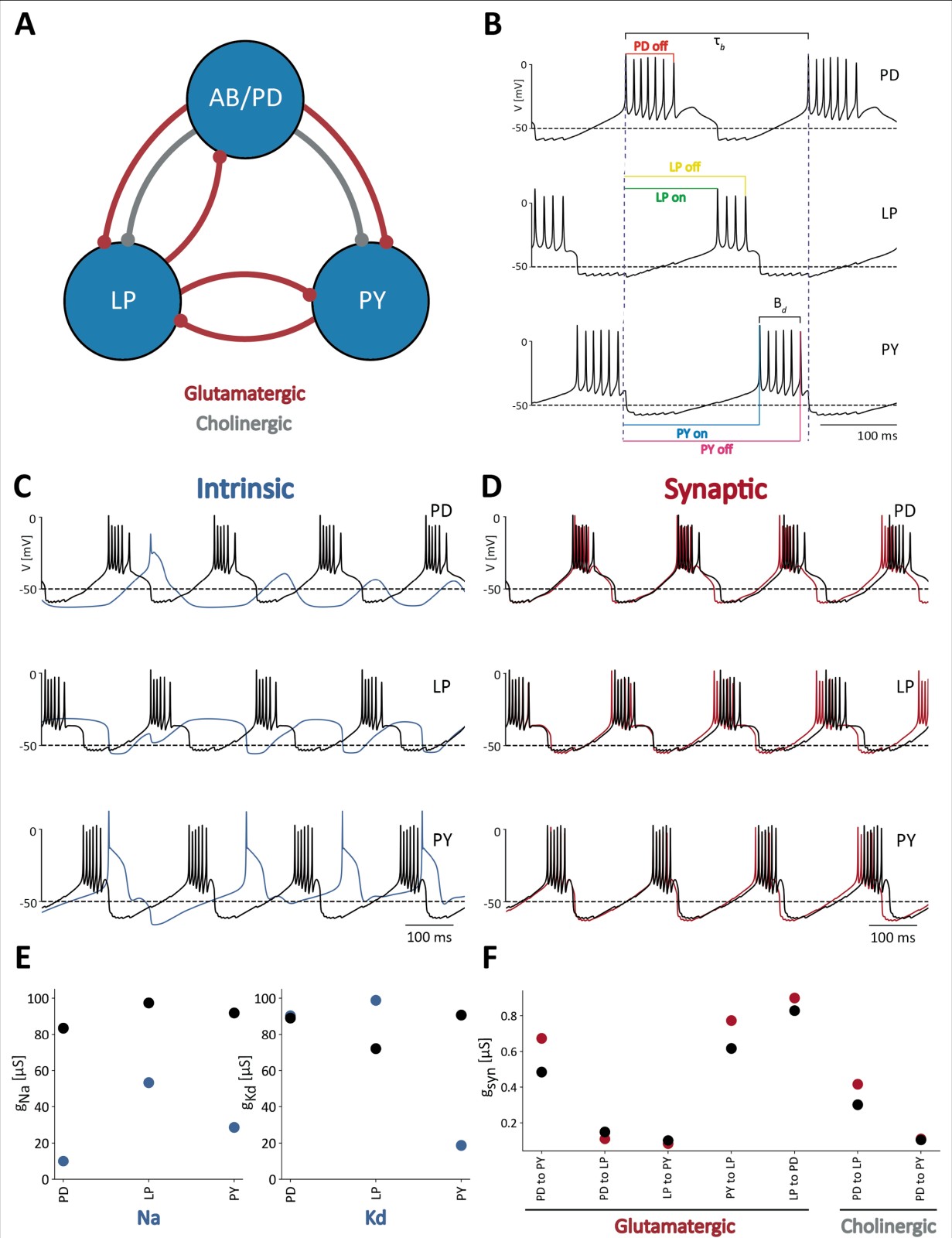

**Figure 1.** Feature extraction to classify model network activity. (**A**) Schematic of the model used in *Prinz et al., 2004*. The three cells are connected by seven inhibitory chemical synapses of two types: the red curves are cholinergic synapses and the grey curves are glutamatergic synapses. (**B**) Pyloric network activity. Each cell displays periodic bursting activity and the cells fire in a sequence *PD-LP-PY*. The color lines indicate the measures used for calculating duty cycle and firing phases. (**C**) Membrane voltages of control network activity (in black), overlaid with the network activity that results

*Figure 1 continued on next page*

*Figure 1 continued*

from changing the intrinsic conductances (blue). (**D**) Membrane voltages of control network activity (in black) overlaid with network activity that results from changing the synaptic conductances (in red). (**E**) Changes in intrinsic maximal conductance values. The black dots show the unperturbed or control conductances and the blue dots correspond to the changed conductances that produced the blue voltage traces in (**C**). Only two maximal conductances per cell are shown for simplicity, but all 9 × 3 = 27 intrinsic conductances were changed. (**F**) Synaptic maximal conductance values for the control network (in black) and the perturbed network (in red) in (**D**).

to the activity of the *same* model when each of the intrinsic conductances are changed by a random amount within a range between 0 (completely removing the conductance) and twice its starting value, $2 \times g_i$, or equivalently, an increment of 100%. In this case, altering the intrinsic conductances results in a pattern of activity that is different from the control activity and is not 'pyloric' because the cells fail to produce bursts of action potentials. *Figure 1D* shows control activity in black, and activity when the synaptic conductances are changed in red. In this case the pattern is similar to control activity, but there are some differences. Despite only changing the synaptic conductances, the network's activity becomes slower and the phases of the cycle at which cell fires are slightly different. *Figure 1E* shows the control values of the *intrinsic* conductances in black, and the changed values of the conductances in blue used in the simulations in *Figure 1C*. For simplicity, we only show how $\bar{g}_{Na}$ and $\bar{g}_{Kd}$ were changed, but all intrinsic conductances were changed. In the LP cell, the sodium conductance $\bar{g}_{Na}$ decreased while the delayed-rectifier potassium conductance $\bar{g}_{Kd}$ increased, leading to a loss of spiking activity. The sodium maximal conductance $\bar{g}_{Na}$ in the PY cell also decreased but $\bar{g}_{Kd}$ decreased even further, thus maintaining spiking activity. These effects are due to the relative magnitude of $\bar{g}_{Na}$ and $\bar{g}_{Kd}$ in the cell, as these are the two largest maximal conductances. Finally, *Figure 1F* shows the control values of the synaptic maximal conductances in black and the perturbed values in red.

## A model database of degenerate solutions

Conductance-based models of neuronal activity are often employed to test and generate hypotheses about the functioning of individual neurons and/or circuits. One difficulty in working with these models is that tuning them to perform specific tasks can be challenging. Finding values of the parameters, such as the maximal conductances $\bar{g}_i$, so that a given circuit will display a specific set of behaviors has been done in the past by painstakingly hand-tuning the models' parameters *Traub et al., 1991*; *Nadim et al., 1995*; *Buchholtz et al., 1992*. One alternative to hand-tuning consists of evaluating all possible combinations of parameter values within a specified domain, building a model database, and filtering the solutions of the models that satisfy a set of criteria *Prinz et al., 2003b*; *Prinz et al., 2004*; *Crasto, 2007*. Another alternative was introduced recently in *Alonso and Marder, 2019*. For any given set of parameters, it is possible to simulate the model, and score the solutions depending on their features. The approach consists of defining a *cost function* that takes values of conductances as inputs, and returns a score. The goal then is to define a cost function such that low scores correspond to solutions of the model that display the target or desired behavior. Thus, *training* the model corresponds to optimizing—finding local minima—of the cost function. This approach was employed successfully to produce models of single neurons and pyloric networks that can operate robustly over a range of temperatures *Alonso and Marder, 2020*. In this study, we followed the approach in *Alonso and Marder, 2020*. We modified the model in *Prinz et al., 2004* to include a modulatory current $I_{MI}$ and we also modified the *cost function* in *Alonso and Marder, 2020* to enforce a greater voltage separation between slow oscillations and spikes (see Methods). In this work, optimization of the *cost function* yields values of conductances such that the circuit displays a pyloric rhythm as in *Figure 1B*. We used a genetic algorithm to optimize the cost function, and in this way generated a database of $N = 100$ models with different values of maximal conductances *Goldberg and Holland, 1988*. All models in the database display pyloric rhythms whose features—such as frequency and burst duration—are consistent with experimental measures performed in the pyloric circuit in the crustacean STG *Hamood et al., 2015*.

The values of the conductances in our model database are widely distributed over many-fold ranges. For example, *Figures 2A and B* show the distributions of the PD A-current conductances and the PD to PY glutamatergic synaptic conductances. These distributions span a broad range and are not normally distributed ($p < 0.05$, Shapiro-Wilk test *Shapiro and Wilk, 1965*). In this study, each model is fully specified by its set of maximal conductances $\bar{g}_i$, while all other parameters such as

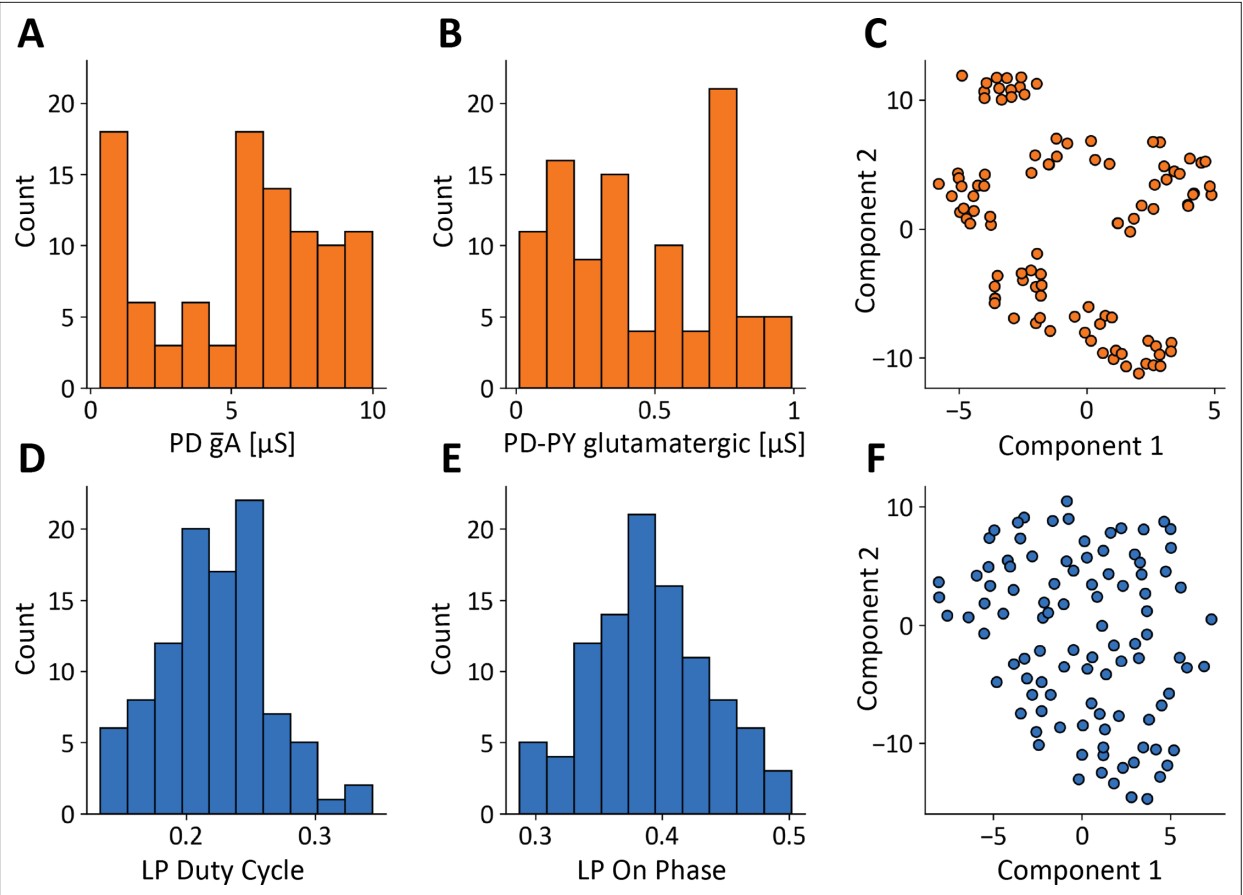

**Figure 2.** A model database of degenerate solutions. (**A, B**) Distributions of $gA$ maximal conductance in PD cells and $PD - PY_{glut}$ glutamatergic synapse maximal conductance ($N = 100$). (**C**) Dimensionality reduction (t-SNE) and visualization of models' maximal conductances. (**D, E**) Distributions of features: LP duty cycle and LP-ON phase. (**F**) Dimensionality reduction (t-SNE) and visualization of models' conductances models' features.

reversal potentials, activation functions, membrane capacitance, were kept fixed and are the same across models. The total number of conductances in a model is $N_g = 34$. To try to visualize the conductances in the database, we employed a popular technique (t-SNE) to visualize high-dimensional vectors. **Figure 2C** shows a t-SNE plot of the model conductances, where we reduced each models' conductance set for visualization in a two-dimensional space **Hinton, 2002**. The clusters in this plot indicate that there are groups of models with similar underlying conductance relationships in the model database.

The models in the database were constrained to produce solutions whose features were similar to the experimental measurements in the STG, but their activities are not identical. For example, **Figure 2D and E** show the distributions of two features: the duty cycle of the LP cell, and the phase of the cycle at which the LP cell begins to fire. Interestingly, these features are normally distributed ($p > 0.05$, Shapiro-Wilk test). Similarly to the values of the conductances, the *features* we extracted from the activity also can be matched to a high dimensional vector with a total number of features of $N_F = 11$. **Figure 2F** shows a t-SNE plot of the models' features. Because the feature vectors are similar, their t-SNE projections do not form groups or clusters.

## Model networks with similar behavior and different underlying currents

Computational and experimental studies suggest that neurons can produce similar activities employing different contributions of each ionic current. This observation is known as *degeneracy* and one way to illustrate this was introduced recently in **Alonso and Marder, 2019**. A currentscape is a representation of neural dynamics that displays the percent contribution of each ionic current to the total inward and outward currents over time. **Figure 3** shows the currentscapes for the PD, LP, and PY neurons in

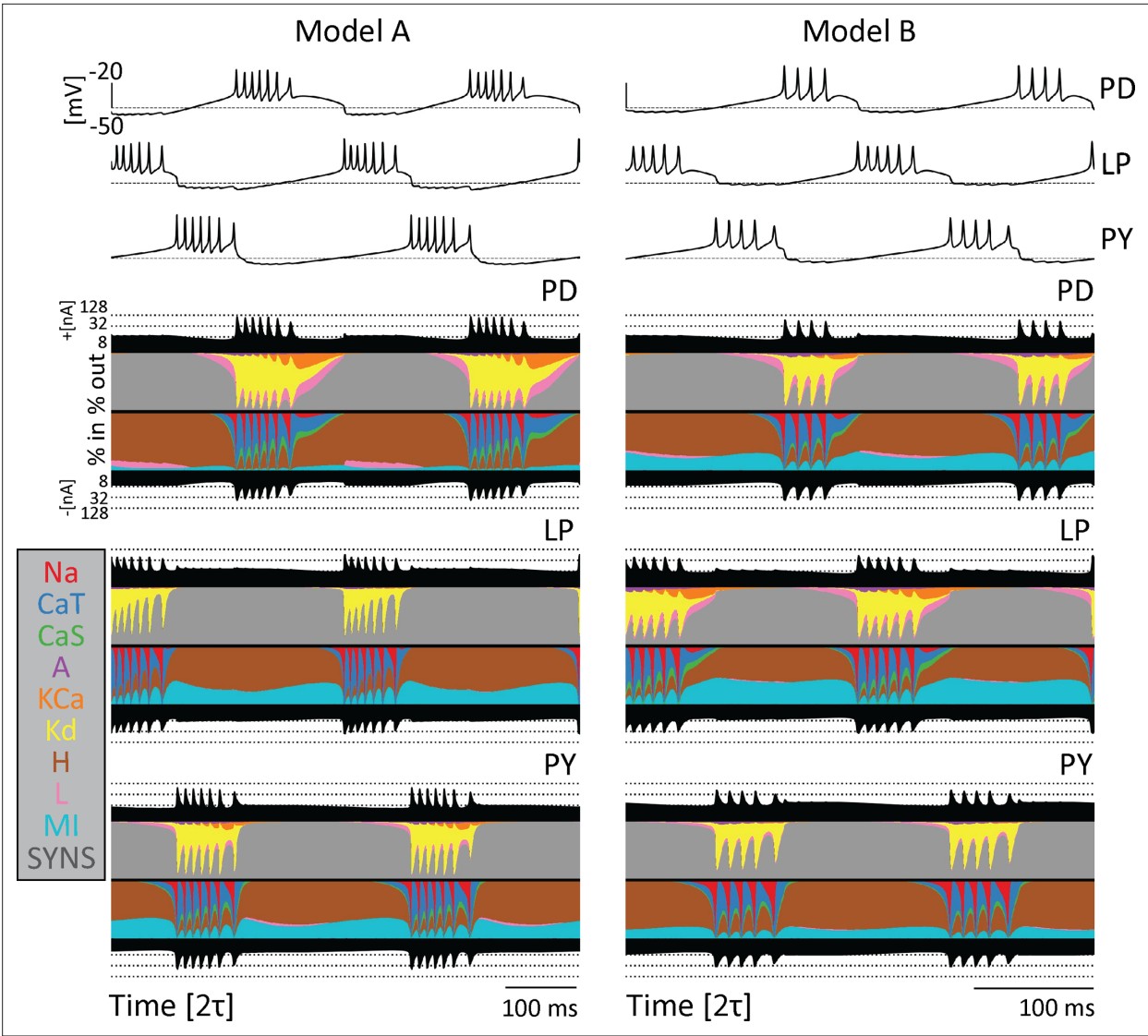

**Figure 3.** Model networks produce similar behavior with different underlying currents. Currentscapes for two different models. The voltage traces for both networks are plotted at the top. The filled curves on top and bottom of the currentscapes for each cell type show the total inward (outward) current over time in logarithmic scale. The colored filled curves indicate the percent contribution of each current over time.

two different models. The voltage traces of each network are plotted above the currentscapes. The total outward and inward current of each cell are shown as black-filled curves above and below each currentscape plot on a logarithmic scale. The dashed lines correspond to reference current values. The percent contributions of each ionic current are represented in colors. The outward currents are on top and inward currents on the bottom.

The currentscapes show visible differences in each cell within a model as well as across the two different models, yet the activity of the networks is similar. For example, there is a visibly larger contribution of the $I_{MI}$ current to the PD cell in Model B as compared to Model A. Likewise, the contribution of the $I_{Kd}$ current is more prominent in Model B's LP cell than Model A's LP cell. In addition, there is little to no contribution of the KCa current in the Model A PY cell, while contributions of this current are more visible in the Model B PY cell. Furthermore, there is a small contribution of the $I_{CaS}$ current in the Model A PD cell, but a larger contribution in the Model B PD cell. Because the activity is driven by different combinations of currents, the excitability properties of the cells are different, and this in turn results in differential sensitivities to perturbations.

## Model networks are more sensitive to perturbation of intrinsic than synaptic conductances

We compared the robustness of intrinsic and synaptic conductances by modifying these two sets separately. For this, we independently varied the values of each maximal conductance $\bar{g}_i$ by some random amount within a range *Giovannini et al., 2013*; *Zang and Marder, 2023*. For each conductance, this variation range is defined as a *percent* deviation from its starting or control value. The larger this percentage, the larger the change in maximal conductances. The values of the modified conductances are obtained by sampling from a uniform distribution over each range. When the range is small and there is a small percent deviation from the control values, the expectation is that the activity will change very little and stay pyloric in most cases. As the variation range is increased, corresponding to larger percent deviations from control values, it is reasonable to expect that the activity will be disrupted in a greater proportion of cases.

We explored changes to maximal conductances between 0% and 100%. The change size or variation range was defined as $\delta \in [0, 1]$. For each conductance, we perform the change $\bar{g}_i \to \bar{g}_i * \gamma$ with $\gamma = 1 + \delta U[-1, 1]$, a random number drawn from a uniform distribution and scaled by $\delta$. In this way, values of $\delta \approx 0$ correspond to small percent changes and the limit case $\delta = 1$ allows for changes of 100% of the value of a given conductance. For each value of $\delta$ we sampled $N = 1000$ sets of conductances. For each set of conductances, we simulated the model, extracted features, and classified the activity as *pyloric* or *not pyloric* using a random forest classifier *Ho, 1995*. We then computed the percentage of traces that displayed pyloric activity and evaluated how this percentage changes as the variation range $\delta$ increases. This procedure was performed independently for both intrinsic and synaptic conductances, allowing us to compare the sensitivity to changes of each set. We repeated the procedure for all one hundred models in our database.

*Figure 4* shows the result of this analysis on ten models. In all panels, the x-axis corresponds to the variation range $\delta$. The y-axis indicates the fraction of traces that were classified as pyloric activity. By definition, when the variation range is zero ($\delta = 0$), all 'perturbations' result in the same model, and therefore, the activity is the same as in control. As expected, the fraction of traces that display pyloric activity decreases as the variation range ($\delta$) increases. This is true for both intrinsic (blue) and synaptic conductances (red). The synaptic sensitivity curve is above the intrinsic sensitivity curve for all values of the variation range $\delta$. This means that the models are more resilient to changes in their synaptic conductances.

To quantify the observation that the intrinsic conductances are more sensitive to perturbations than the synaptic conductances, we calculated the average sensitivity curves in *Figure 5A*. The plot shows the average fraction of pyloric traces across all models for all values of the variation range $\delta$. To further quantify these observations, we fitted each sensitivity curve with a sigmoid function (*Figure 5B*), $S(\delta) = c \cdot \left(1 + \frac{1}{1+e^{-(\delta-b)/a}}\right)$, that best approximates it. The parameters are $a$, which controls the width, and $b$, which determines the midpoint. Parameter $c$ controls the overall amplitude of the sigmoid and was kept fixed at $c = 100$, since by definition it must be $S(0) = 100$.

The sigmoid functions approximate the sensitivity curves well, with an average correlation coefficient of $r^2 = 0.98$, and an average error of approximately 1% for each value of $\delta$. The fits to the synaptic sensitivity curves were marginally better than in the intrinsic case as the mean error for the synaptic was 0.55% ($r^2 = 0.98$) , whereas it was 1.54% ($r^2 = 0.98$) for the intrinsic.

To compare the intrinsic and synaptic sensitivity curves, we inspected the distributions of the sigmoidal fit parameters (*Figure 5C and D*). These sets of parameters were not normally distributed ($p < 0.05$, Shapiro-Wilk test) except for the midpoint parameter in the intrinsic curves ($p = 0.13$). To evaluate if two distributions were statistically different, we used *Student, 1908*. The midpoint parameter distributions are different, and there is little to no overlap between them ($p < 0.001$). In addition, we found that the intrinsic conductances have much smaller midpoint parameters, consistent with the observation that their sensitivity curves decrease faster as the size of the conductance changes $\delta$ increases.

Although the distributions of midpoint parameters are visibly different, the width parameter distributions largely overlap. However, these distributions are significantly different ($p < 0.001$, Student's t-Test) because the red distribution is long-tailed. A smaller midpoint parameter and a smaller width parameter both indicate that a curve decreases more quickly. This difference in the midpoint parameters indicates that the synaptic conductances must be varied further from their starting values to disrupt pyloric activity.

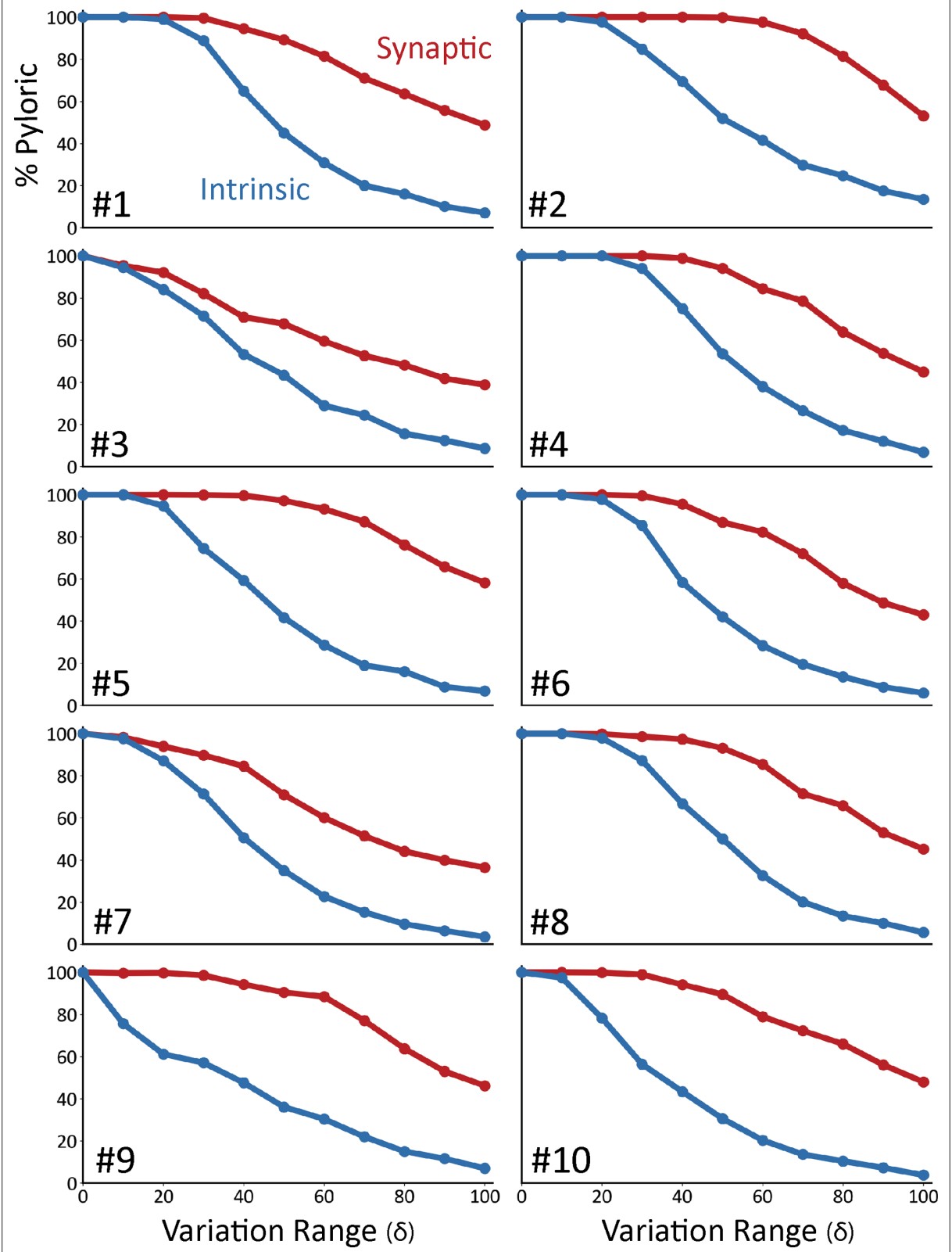

**Figure 4.** Model networks are differentially sensitive to modification of intrinsic and synaptic conductances. Curves representing the decrease in the percentage of pyloric models as a function of the variation range. Curves from intrinsic conductance perturbations are shown in blue, and synaptic in red. Ten sets of curves are shown for 10 models with different combinations of maximal conductances.

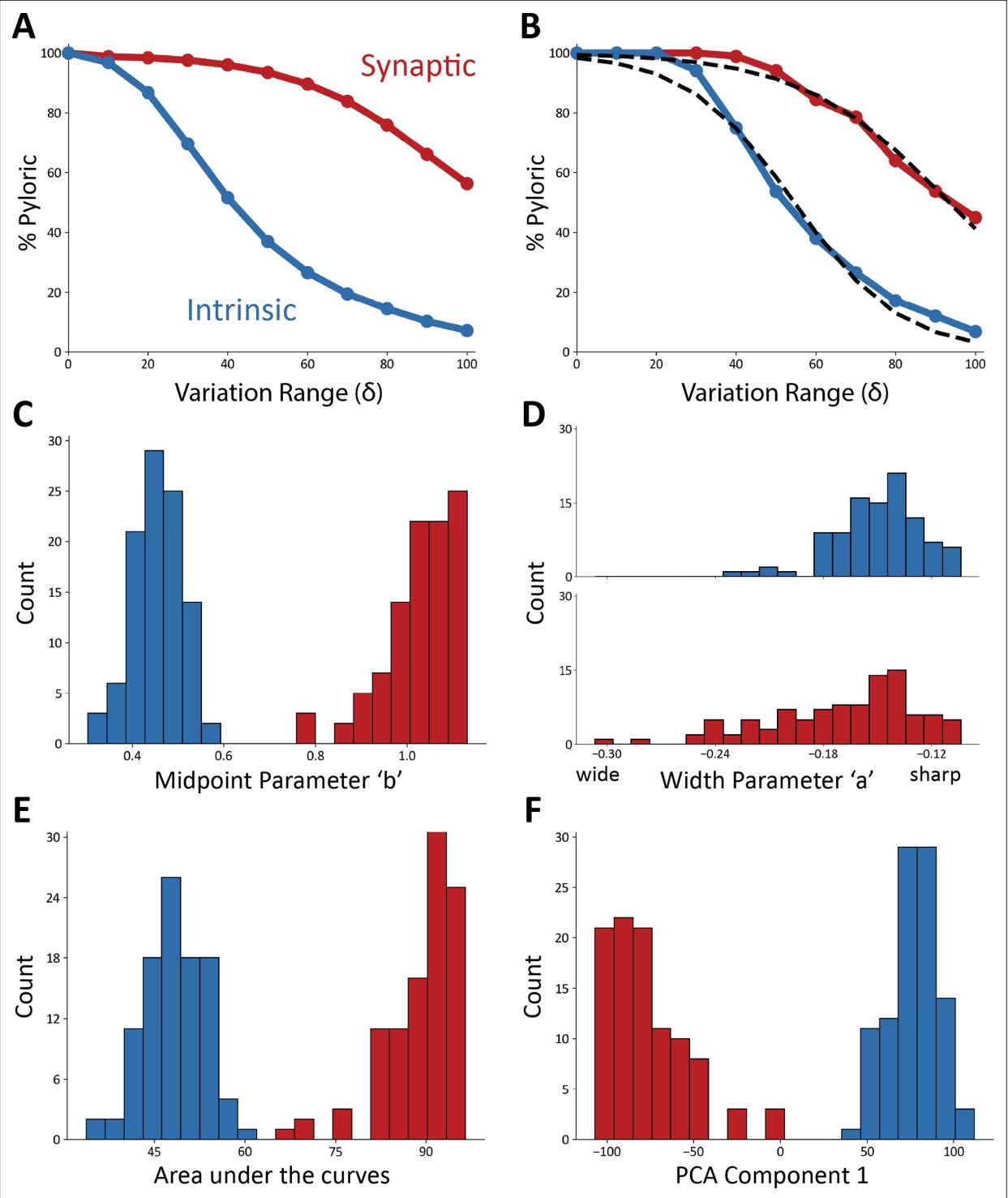

**Figure 5.** Model networks are more sensitive to changes in intrinsic conductances. (**A**) Average sensitivity curves across models. The plots show the average percentage of pyloric models for each value of δ (intrinsic curves show in blue, synaptic in red). (**B**) Example sigmoidal fits for one set of intrinsic and synaptic sensitivity curves. (**C**) Distributions of midpoint parameters for all fits of the sensitivity curves (intrinsic in blue, synaptic in red, 20 bins). (**D**) Distributions of width parameters for all fits of the sensitivity curves (intrinsic in blue, synaptic in red, 20 bins). (**E**) Distributions of the areas under the curves (intrinsic in blue, synaptic in red, 20 bins). (**F**) Distribution of eigenvalues for the first PCA component (97.6%; intrinsic in blue, synaptic in red, 20 bins).

To further compare the intrinsic and synaptic sensitivity curves, we calculated the areas under the curves (*Figure 5E*). The areas under the synaptic curves are significantly larger ($p < 0.001$, Student's t-Test), consistent with the previous observation that the network is more robust to perturbations in their synaptic conductances.

To characterize the variability in the sensitivity curves, we performed principal component analysis (PCA). The first component captures 97.6% of the variance, which indicates that the curves are similar and predominantly vary along a single dimension. The distribution of the eigenvalues required to reconstruct the curves from the principal component are shown in *Figure 5F*. The eigenvalues for the intrinsic and synaptic curves result in visibly different distributions ($p < 0.05$, Student's t-Test) with means of opposite signs. These eigenvalues are normally distributed in the case of the intrinsic curves ($p = 0.82$, Shapiro-Wilk test), and not normally distributed in the synaptic case ($p < 0.05$). The intrinsic and synaptic curves can be reasonably approximated by the same sigmoid-looking curve, but flipping the sign.

## Relative robustness of circuit models visualized in conductance space

These three neuron models require the specification of $N_i = 27$ intrinsic conductances and $N_s = 7$ synaptic conductances. There are multiple combinations of $\bar{g}_i$ that result in a proper pyloric rhythm. This means that there are regions in conductance space $G$ for which the activity is pyloric, $G_P$. The shape and properties of $G_P$ have been explored and studied in the past *Taylor et al., 2006*. To gain some intuition about the properties of these regions, we exploited the trained classifier and evaluated its output over 2D ranges of conductance space.

*Figure 6* shows 2D projections of the regions of conductance space for which the activity is pyloric $G_P$, for two different models. The top rows show the projection of $G_P$ over pairs of intrinsic conductances, $\bar{g}_{Na}$ and $\bar{g}_{Kd}$. The blue regions correspond to values of the conductances for which the activity is pyloric ($G_P$), and the white cross indicates the 'control' or initial values of the conductances for that model. The white box indicates the maximum range of the perturbations used in the sensitivity analyses ($\delta = 1$).

The shape of $G_P$ is intricate and complex. Despite this, some basic properties of $G_P$ are revealed. The top panels of *Figure 6* show the projection of $G_P$ over the same pair of intrinsic conductances, across cell types. The blue regions across panels are different, indicating that the same perturbation in different cell types may result in drastically different outputs: cell identity matters. The white cross is closer to the edge in the case of the LP cell, meaning that this particular model is more sensitive to perturbations in the LP cell than to perturbations in the other cells.

The second row in *Figure 6*, model A, shows the projection of $G_P$ over pairs of *synaptic* conductances. The red areas correspond to values of the conductances for which the activity is pyloric ($G_P$), and the white cross indicates the starting or 'control' values of this particular model. As before the shape of these projections are intricate and complex. They also are different across pairs of conductances. Notice that with the exception of the cholinergic synapses (first panel), the initial value of the conductances (white cross) is farther away from the edge than in the intrinsic cases, consistent with the finding that the model is more robust to perturbations in its synaptic conductances. In the same vein, note that the proportion of area of the white box ($\delta = 1$) occupied by the red (synaptic) region is visibly larger than the proportion of area occupied by the blue (intrinsic) region. We computed the proportions of areas of the white boxes that correspond to pyloric activity. These values for the intrinsic conductances panels are $PD = 0.58$, $LP = 0.50$, $PY = 0.49$, and the proportions for the synaptic conductances panels are $PD_{PY} = 0.62$, $PD_{LP} = 0.87$, and $LP_{PD} = 0.94$. The occupied areas for synaptic conductances are larger than in the intrinsic conductances panels, consistent with our finding that the circuits' activities are more robust to changes in synaptic conductances than changes in intrinsic conductances.

*Figure 6* (bottom) shows the same calculation in a different model (model B). As before, the blue regions and the red regions are different across cells and pairs of synapses. The shape of the projections of $G_P$ are different across models. In the case of the intrinsic conductances the shapes are similar overall, but there are visible differences, such as the white cross in the LP cell being farther away from the edge, as in model A. The projection of $G_P$ on pairs of synaptic conductances is very different in this model. In this case, almost the totality of the area enclosed by the white boxes ($\delta = 1$) corresponds to $G_P$, suggesting that model B is more robust than model A to changes in synaptic conductances.

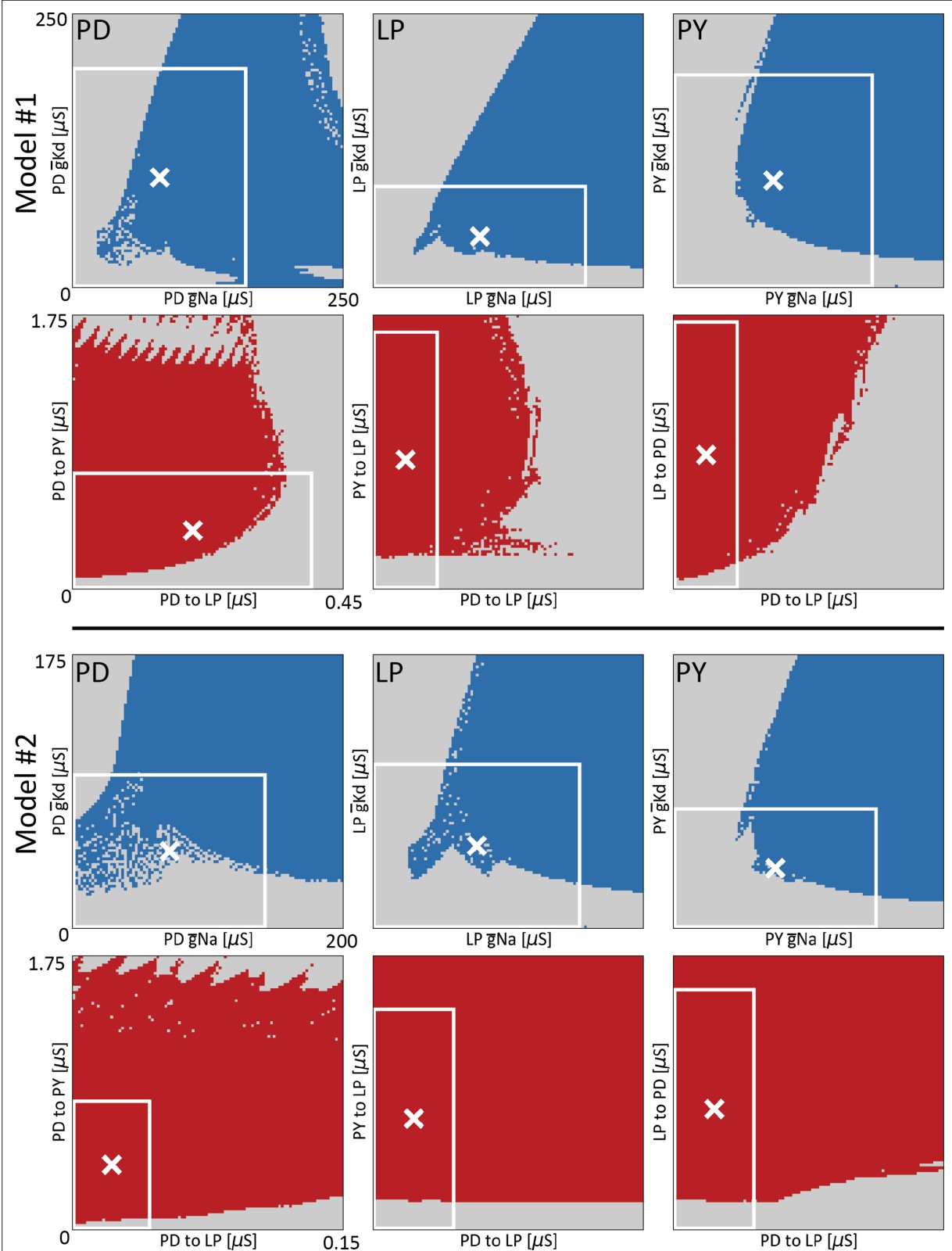

**Figure 6.** Relative robustness of models visualized in conductance space. Ranges of conductance space where the activity of the network is pyloric, for two different models (model #1 and model #2). All panels show the output of the classifier over 2D ranges of conductance space. The top panels show pairs of intrinsic maximal conductances ($\bar{g}_{Na}$ and $\bar{g}_{Kd}$ of each cell). The blue dots correspond to values of the conductances that produce pyloric activity, while the gray dots indicate values were the network activity is not pyloric. The bottom panels show pairs of synaptic maximal conductances for

*Figure 6 continued on next page*

*Figure 6 continued*

which the activity is pyloric in red. The white cross indicates the values of the conductances for the models. The white boxes indicate the extent of the perturbations when they are allowed to be as large as a 100% deviation from their control values ($\delta = 1$).

This is consistent with the finding that the synaptic sensitivity curve is steeper in model A (Model #1 in *Figure 4*) than in model B (Model #2 in *Figure 4*). As before, we computed the proportion of areas of pyloric activity within the white boxes: $PD = 0.61$, $LP = 0.55$, $PY = 0.52$, and the proportions for the synaptic conductances panels are $PD_{PY} = 0.88$, $PD_{LP} = 0.87$, and $LP_{PD} = 0.83$. These results provide an intuition of the complexities of $G_P$. Not only are these regions hard-to-impossible to characterize in one circuit, but they are also different across circuits.

## Discussion

A major goal of much of neuroscience is to understand how experience alters the function of brain circuits, both in the context of development and learning. At the same time, we are aware that damage to the brain can produce altered brain structure. It is therefore important to understand the resilience of brain circuits to damage, and to understand the plasticity mechanisms that foster learning and recovery from damage. Thus, it is critical to understand which changes in circuit parameters are most influential in altering circuit function.

Despite the enormous body of work on the importance of alterations of synaptic strength in numerous preparations, there are also many well-documented cases of neurons that change their intrinsic excitability during development or as a function of neuromodulation. For example the Rohan-Beard cells first show a slow $Ca^{2+}$- dependent spike, before eventually acquiring a rapid $Na^+$- dependent action potential *Baccaglini and Spitzer, 1977* and ACh and biogenic amine neuromodulators directly alter the intrinsic properties of thalamic and other neurons *McCormick and Pape, 1990*; *McCormick et al., 1991*; *McCormick and Prince, 1986*; *McCormick and Prince, 1987*; *McCormick and Prince, 1988*; *McCormick and Wang, 1991*; *Williams and Fletcher, 2019*. There are myriad other examples in which neuromodulators completely transform the dynamics of neuronal activity *Bargmann, 2012*; *Marder, 2012*; *Nusbaum and Blitz, 2012*. A neuromodulator that changes a neuron's activity profile from silent, to tonic firing, to producing plateau potentials or oscillatory bursts of activity is likely to appreciably alter the activity of the circuits in which these neurons are found. Because altering an intrinsic current is often accompanied by changes in the time scales of neuronal activity, even modest changes in an intrinsic conductance can result in major alterations in the dynamics of a target neuron and the circuits in which it functions *Desai et al., 1999a*, *Desai et al., 1999b*. In a sense, we biased the results of this study in manner that underestimates the influence of intrinsic activity by selecting for neurons that were within a restricted region of the degenerate range of solutions.

We expect that results similar to these will be found in many brain regions that show strong propensities for plateau and oscillatory properties, especially in response to neuromodulation. It remains to be seen whether cortical or hippocampal regions, typically thought to be modified by changes in synaptic strength *Hasselmo et al., 1995*, are less sensitive to modulation of intrinsic properties.

In this study, we evaluated the role of changes in inhibitory synaptic strengths, as in the STG synaptic connections are inhibitory. Because increases in inhibitory synaptic conductances push the membrane potential closer to reversal potential, there can be a diminishing effect of increasing the inhibitory synaptic strength, although the increase in the post-synaptic conductance still decreases the efficacy of all other currents, and changes in inhibitory synaptic currents can saturate *Prinz et al., 2003a*.

Although increasing the strength of excitatory synaptic currents also is somewhat diminished as the membrane potential of neurons moves closer to the reversal potential of an EPSP, the effect of increasing the EPSP may be amplified as voltage dependent currents are activated. Therefore, it is possible that a circuit dominated by excitation may be more dominated by changes in synaptic than intrinsic currents. Nonetheless, a large body of work now argues that much cortical function depends on a balance of excitatory/inhibitory function *van Vreeswijk and Sompolinsky, 1996*; *Atallah and Scanziani, 2009*; *Isaacson and Scanziani, 2011*; *Lim and Goldman, 2013*; *Lo et al., 2015*; *Rubin et al., 2017*; *Wang, 2020*; *Sadeh and Clopath, 2021*, so a similar study such as this needs to be done in a modeled cortical or hippocampal circuit. That said, this study benefits greatly from a simple

measure of circuit performance that can be used to determine circuit robustness, and this is less obvious in many other circuits. We believe our results will hold for other rhythmic circuits and will be relevant for similar studies in other circuits with more complex functions.

# Methods

## The Model

We employed a model circuit inspired by the STG of crustaceans that was developed and studied previously by multiple authors *Golowasch et al., 1999*; *Prinz et al., 2004*; *Alonso and Marder, 2020*. The model network is composed of three cells named PD, LP, and PY, each modeled by a single compartment with nine currents: all cells have a sodium current, $I_{Na}$; a delayed rectifier potassium current, $I_{Kd}$; transient and slower calcium currents, $I_{CaT}$ and $I_{CaS}$; a transient potassium current, $I_A$; a calcium-dependent potassium current, $I_{KCa}$; a hyperpolarization-activated inward current, $I_H$; a leak current, $I_{leak}$; and a neuromodulatory current $I_{MI}$ *Golowasch and Marder, 1992b*; *Swensen and Marder, 2000*. This model of the pyloric network is identical to previous ones, except for the inclusion of the neuromodulatory current $I_{MI}$.

The membrane potential of the cells evolves according to a Hodgkin-Huxley equation for a single compartment,

$$C\frac{dV}{dt} = I_e - \sum_{i=1}^{9} I_i. \tag{1}$$

The terms in the sum are the ionic currents $I_i = \bar{g}_i m_i^{p_i} h_i^{q_i}(V - E_i)$ and $I_e$ is injected current. The maximal conductance of a channel is given by $\bar{g}_i$. The variables $m_i$ and $h_i$ correspond to the activation and inactivation processes. The integers $p_i$ and $q_i$ are the number of gates in each channel. The reversal potential of the ionic current associated with the $i - th$ current is given by $E_i$.

The reversal potential of the Na, K, H, leak, and $I_{MI}$ currents were kept fixed at $E_{Na} = 30mV$, $E_K = -80mV$, $E_H = -20mV$, $E_{leak} = -50mV$, and $E_{IMI} = -10mV$. The reversal potential of the calcium currents $E_{Ca}$ was updated using the Nernst equation assuming an extracellular calcium concentration of $3 \times 10^3 \mu M$. The kinetic equations describing the seven voltage-gated conductances were modeled as in *Liu et al., 1998*,

$$\tau_{m_i}(V)\frac{dm_i}{dt} = m_{\infty_i}(V) - m_i$$
$$\tau_{h_i}(V)\frac{dh_i}{dt} = h_{\infty_i}(V) - h_i. \tag{2}$$

The functions $\tau_{m_i}(V)$, $m_{\infty_i}(V)$, $\tau_{h_i}(V)$, and $h_{\infty_i}(V)$ are based on the experimental work of *Turrigiano et al., 1995*. The activation functions of the $K_{Ca}$ current require a measure of the internal calcium concentration $[Ca^{+2}]$ (*Liu et al., 1998*). The dynamics of the intracellular calcium concentration are given by,

$$\tau_{Ca}\frac{d[Ca^{+2}]}{dt} = -Ca_F(I_{CaT} + I_{CaS}) - [Ca^{+2}] + Ca_0. \tag{3}$$

Here, $Ca_F = 0.94 \frac{\mu M}{nA}$ is a current-to-concentration factor and $Ca_0 = 0.05 \,\mu M$. The total capacitance of the cell is $C = 1nF$.

The interactions in the network consist of seven chemical synapses and are similar to *Prinz et al., 2004*. The synaptic current is given by $I_s = \bar{g}_s s(V_{post} - E_s)$, where $\bar{g}_s$ is the synaptic strength, $V_{post}$ is the membrane potential of the postsynaptic neuron and $E_s$ is the reversal potential of the synapse. The activation of a synapse $s(t)$ is given by

$$\frac{ds}{dt} = \frac{s_{\infty}(V_{pre}) - s}{\tau_r + \tau_s} \tag{4}$$

with,

$$s_\infty(V_{pre}) = \frac{1}{1 + exp((V_{th} - V_{pre}/\Delta))}, \tag{5}$$

and

$$\tau_s = \frac{1 - s_\infty(V_{pre})}{k_-}. \tag{6}$$

These equations are identical to *Prinz et al., 2004* except for the inclusion of a bound for the timescale of activation $\tau_r = 20msec$ (*Alonso and Marder, 2020*).

## Generating the model database

We generated 100 different models of the pyloric network following the approach outlined in *Alonso and Marder, 2019*; *Alonso and Marder, 2020*, which we briefly review here. Each model is specified by a set of maximal conductances, while all other parameters (such as reversal potentials) are the same across models. Our approach consists of defining a *target* or *objective* function such that upon minimization, it will yield sets of conductances that result in normal or control pyloric activity.

The *target function* takes values of conductances as its input. The model is integrated for 20 s and the first 10 s are dropped to minimize transient activity. We then compute several quantities: the average ($<>$) burst frequency $< f_b >$, the average duty cycle $< dc >$, the number of crossings with a slow wave threshold $\#_{sw} = -50mV$, the number of bursts $\#_b$, and the average lags between bursts $< \Delta_{PD-LP} >$ and $< \Delta_{PD-PY} >$. In this study, we also sought solutions that would display a large voltage separation between the slow wave and the spike troughs. This was done by defining a second *slow wave threshold* at $\#_{sw} = -45mV$ and checking that the number of crossings is the same as the number of bursts. Finally, to penalize unstable solutions we compute the standard deviation of the burst frequency and duty cycle distributions; a solution was penalized if $std(\{f_b\}) \geq < f_b > \times 0.1$ or $std(\{dc\}) \geq < dc > \times 0.2$. If a solution is not discarded, we can use these quantities to score its similarity to the target, pyloric behavior. We define the *errors* as follows,

$$E_f = \sum_{i=\text{cell}} (f_{tg} - < f_b >_i)^2 \tag{7}$$

$$E_{dc} = \sum_{i=\text{cell}} (dc_{tg} - < dc >_i)^2$$

$$E_{sw1} = \sum_{i=\text{cell}} (\#_{sw} - \#_b)^2$$

$$E_{sw2} = \sum_{i=\text{cell}} (\#_{sw} - \#_b)^2$$

$$E_{ph} = (\Delta_{PD-LPtg} - \frac{< \Delta_{PD-LP} >}{\tau_b})^2 + (\Delta_{PD-PYtg} - \frac{< \Delta_{PD-PY} >}{\tau_b})^2. \tag{8}$$

$E_f$ measures how far the bursting frequency of each cell is to a target frequency $f_{tg} = 3$, $E_{dc}$ accounts for the duty cycle, $E_{sw}$ accounts for the difference between the number of bursts and the number of crossings with the slow wave threshold at $t_{sw} = -50mV$ (if $\#_{sw} \neq \#_b$ then $E_{sw}$ is large), $E_{sw2}$ is the same as $E_{sw}$ but for a different threshold value ($t_{sw2} = -45mV$), $E_{ph}$ compares the lags between bursts (in units of the bursting period $\tau_b = \frac{1}{<f_b>}$) to a target lag $\Delta_{tg}$. These measures are discussed in more detail in *Alonso and Marder, 2019*; *Alonso and Marder, 2020*.

For each set of maximal conductances $\bar{g_i}$ we can then define an objective function,

$$E(\mathbf{G}) = \alpha E_f + \beta E_{dc} + \gamma E_{sw} + +\gamma_2 E_{sw2} + \eta E_{ph}. \tag{9}$$

The coefficients ($\alpha, \beta, \gamma, \gamma_2, \eta$) determine which features are more important. In this work we used $\alpha = 10$, $\beta = 1000$, $\gamma = 1$, $\gamma_2 = 1$, $\eta = 10$. We found that using these weights results in a *target function* that produces useful sets of conductances (that produce normal pyloric triphasic activity). All penalties $E_i$ were calculated using $T = 10$ secs. The simulations were done using an *RK4* integration routine with $dt = 0.05$ msecs.

The *target control behavior* is defined as all cells bursting with 20% duty cycle for *PD* ($dc_{tg_{PD}} = 0.2$) and 25% for the LP and PY cells ($dc_{tg_{LP,PY}} = 0.25$). The lag between bursts was targeted to be $\Delta_{PD-LPtg} = 0.5$ and $\Delta_{PD-PYtg} = 0.75$. The target burst frequency of all cells was set to $f_{tg} = 1Hz$ (*Bucher et al., 2006*;

*Tang et al., 2010*; *Hamood et al., 2015*). Minimization of the *target function* successfully produces sets of maximal conductances $g_i$ that produce normal or control triphasic rhythms.

We defined a search space of allowed values listed here: for each cell we searched for $\bar{g}_{Na} \in [0, 10^2]$ ($[\mu S]$), $\bar{g}_{CaT} \in [0, 10^1]$, $\bar{g}_{CaS} \in [0, 10^1]$, $\bar{g}_A \in [0, 10^1]$, $\bar{g}_{KCa} \in [0, 10^1]$, $\bar{g}_{Kd} \in [0, 10^2]$, $\bar{g}_H \in [0, 10^1]$, $\bar{g}_L \in [0, 1]$, $\bar{g}_{MI} \in [0, 10]$. All synaptic conductances were searched in the range $\bar{g}_{syn} \in [0, 1]$ ($[\mu S]$). We discretized the search space by taking $10^3$ equally spaced values for each parameter. The objective function (*Equation 9*) was minimized using a custom genetic algorithm *Holland, 1992* on a desktop computer with 32 cores. We ran the genetic algorithm using 1000 initial individuals taken at random in the search space for 300 generations. Finding a successful population of models takes several hours. We repeated this process to produce 100 different models with values of conductances distributed over large ranges.

## Extracting features from model traces

The models were simulated for 10 s, and the final 5 s of the simulation were utilized. We calculated spike times by finding the local maxima of the time series of the membrane potential of each cell. Some subthreshold depolarizations resulted in local maxima that were not spikes. To isolate the spikes from these subthreshold depolarizations, local maxima were filtered to only include spike times within 5ms of the local maxima of the derivative of the voltage $\frac{dV}{dt}$. This is because local maxima of the voltage derivative will typically be within 5ms of the local maxima of the voltage if the rise in voltage is very rapid, which is the case for spikes produced by rapid sodium activation. In addition, we computed the maximum value of voltage in the 5 s of simulation that we analyzed ($V_{max}$), and we only kept spikes that were less than $25mV$ below this value. To prevent spike times from being detected in non-firing cells with fast but small oscillations, we imposed that if $V_{max} < -25mV$, then the spike threshold was set to $-25mV$. Models with cells that fire spikes beneath $-25mV$ do exist but were not included in this work.

To calculate the bursts' start and end times, we set a threshold to distinguish inter-burst intervals (IBIs) from inter-spike intervals (ISIs) $ISI_{th}$. In the case of a regularly bursting cell, the ISI distribution is bimodal, with one peak corresponding to the intraburst ISI, and the other peak corresponding to the IBIs. We calculated the ISI threshold by first detecting the peaks in the histograms of ISIs and then chose the value halfway between these peaks.

In some cases the distribution of ISIs was not bimodal due to irregular bursting or tonic spiking. This made it difficult to determine an ISI threshold to correctly identify bursts. In cases where the threshold could not be estimated, or it was estimated to be larger than 3oomS or smaller than 100mS, we set $ISI_{th} = 100mS$. This allowed us to reliably calculate phases by counting tonic spikes in the triphasic rhythm as burst starts and burst ends while also detecting spikes with large ISIs at the end of a burst. Using this method led to errors in burst identification in some cases, and this is a source of error and a limitation of the study.

The features extracted from each model include the mean and standard deviations of the distributions of duty cycles that each cell produces over the period of 5 s. We also calculated the means of the distributions of the phases of the rhythm at which each cell begins and ends a burst. We calculated the duty cycles for each cell as the duration of the burst $B_d$ divided by the bursting period ($\tau_b$),

$$\text{DC} = \frac{B_d}{\tau_b} \tag{10}$$

The phases were calculated as the difference between the cell's on/off times and the start of burst in the PD cell $PD_{on}$ (taken as a reference) divided by the period of the rhythm ($\tau_n$),

$$\theta = \frac{cell_{on/off} - PD_{on}}{\tau_n} \tag{11}$$

We defined $\tau_n$ as the bursting period $\tau_b$ of the PD cell, thus when PD either did not fire or fired irregularly phases were not calculated and these quantities were set to 0.

## Classification of model traces

The activity of each trace was evaluated by first building a vector containing features of its activity. The features are: the means of the distributions of duty cycles for each cell, the standard deviations of the distributions of duty cycles for each cell, and the means of the distributions of phases for each

cell. We attempted classification of the vectors using a number of algorithms including random forest classifiers and gradient boosting machines *Ho, 1995*; *Friedman, 2001*, and we chose to use random forest classifiers to avoid the possibility that gradient boosting machines might over fit to the training data for our purposes *Natekin and Knoll, 2013*. A random forest classifier is a machine learning method used to classify items based on a list of the item's features *Ho, 1995*. The classifier we created consisted of 500 decision trees (max depth = 4), where each tree was fed a different subset of the features for classification. A winner-takes-all vote between the trees chooses the output label of the classifier. The classifier must first be trained with a set of data that was manually labeled, with the help of experimentalists that study the pyloric network in the crustacean STG. When the cells were bursting in a triphasic pattern in the order *PD-LP-PY*, with at least two spikes per burst, we considered the activity to be pyloric (see *Figure 1*). We trained the classifier on a database of 1200 model solutions or traces featuring a wide variety of wave-forms. Any other type of activity was considered 'not pyloric'. In some cases we excluded rhythms that displayed pyloric characteristics but also exhibited irregular features, such as abnormally large duty cycles, cells with low amplitude spikes, and inconsistent firing and/or bursting patterns.

We found that the trained classifier generalized with $\geq 95\%$ accuracy for many, but not all models. To verify that the classifier properly classified the models we chose, we manually checked thousands of traces. Only models that were classified with this level of accuracy after perturbation were used in this study. On rare occasions, the classifier misclassified models that are clearly pyloric as not pyloric even though the list of features fed to the classifier were calculated properly. The percentage of misclassifications was approximately $2 - 3\%$, which is consistent with the accuracy of the classifier.

## Sigmoidal fits

To analyze the sensitivity curves quantitatively, we fit them with sigmoid functions. We optimized for the sigmoidal midpoint and width parameters by minimizing an error function using a brute force approach. The sigmoid function is given by,

$$S(\delta) = c \cdot \left( 1 + \frac{1}{1 + e^{-(\delta - b)/a}} \right) \tag{12}$$

where the $a$ parameter controls the width of the sigmoid, the $b$ parameter determines the midpoint of the sigmoid, and the $c$ parameter corresponds to the maximum asymptote of the sigmoid function. Variable $\delta$ represents the variation range parameter. Parameter $c$ was fixed at 100% because this is the maximum percentage of models that can be pyloric, and by definition all curves start at 100% when the size of the perturbation is $\delta = 0$ (i.e. we impose $S(\delta = 0) = 100$). To find the values of $a$ and $b$ that best approximate the sensitivity curves we minimize an error function defined as follows,

$$E(a, b) = \sum_{i=0}^{N_\delta} \left( (S(\delta_i, a, b) - y(\delta_i))^2 \right) \tag{13}$$

where $\delta_i = [0, 0.1, ....1]$ are the values of $\delta$ for which we computed the sensitivity curves ($N_\delta = 11$), and $y(\delta_i)$ are the values of a sensitivity curve for each value of $\delta_i$. We evaluated all possible combinations of the a and b parameters over a 100 × 100 2D grid of possible pairs with (-0.5 to -0.01 for a, 0 to 2 for b), and kept the values of a and b that minimized the error function $E$.

## Conductance ranges

There are regions in conductance space where the activity is a normal pyloric rhythm. Visualizing and characterizing these regions is challenging because they are high dimensional objects, and we are limited to visualizing their two-dimensional projections. We evaluated the output of the classifier over two-dimensional ranges in conductance space. For this we chose a pair of conductances $(\bar{g}_x, \bar{g}_y)$ and picked 100 values between 0 and $\bar{g}_{max}$ for each pair. We then evaluated network activity using the classifier on the $10^4$ possible pairs of $(\bar{g}_x, \bar{g}_y)$ over this range. The values of $(\bar{g}_x, \bar{g}_y)$ that produce pyloric activity were plotted in color (blue, red) and the values of $(\bar{g}_x, \bar{g}_y)$ that result in non-pyloric activity were plotted in grey. For the intrinsic conductances, we chose to visualize the $\bar{g}_{Na}$ and $\bar{g}_{Kd}$ maximal conductances projection for each cell. For the synaptic conductances, we visualized three pairs: the cholinergic inputs to LP and PY from PD, the glutamatergic inputs to LP from PD and PY, and the

glutamatergic inputs from PD to LP and LP to PD. We performed this procedure on two different models. For *Model A*, $\bar{g}_{max}$ for $\bar{g}_{Na}$ and $\bar{g}_{Kd}$ was $250\mu S$. In the case of the synaptic conductances $\bar{g}_{max}$ was set to $0.45\mu S$ on the x-axis and $1.75\mu S$ on the y-axis. For *Model B*, $\bar{g}_{max}$ was $200\mu S$ for $\bar{g}_{Na}$ and $175\mu S$ for $\bar{g}_{Kd}$. $\bar{g}_{max}$ for the synaptic conductances was $0.15\mu S$ on the x-axis and $1.75\mu S$ on the y-axis.

## Additional information

### Funding

| Funder | Grant reference number | Author |
| --- | --- | --- |
| National Institute of Neurological Disorders and Stroke | NS097343 | Eve Marder |
| National Institute of Mental Health | MH046742 | Eve Marder |

The funders had no role in study design, data collection and interpretation, or the decision to submit the work for publication.

### Author contributions

Zachary Fournier, Software, Formal analysis, Investigation, Visualization, Methodology, Writing – original draft; Leandro M Alonso, Resources, Software, Supervision, Validation, Investigation, Methodology, Writing – original draft, Writing – review and editing; Eve Marder, Conceptualization, Resources, Supervision, Funding acquisition, Investigation, Writing – original draft, Project administration, Writing – review and editing

### Author ORCIDs

Zachary Fournier  https://orcid.org/0009-0009-5681-463X
Leandro M Alonso  https://orcid.org/0000-0001-8211-2855
Eve Marder  https://orcid.org/0000-0001-9632-5448

Reviewer #1 (Public review): https://doi.org/10.7554/eLife.102938.3.sa1
Author response https://doi.org/10.7554/eLife.102938.3.sa2

## Additional files

### Supplementary files

MDAR checklist

### Data availability

All code and data in this study can be accessed at GitHub (copy archived at *Fournier, 2024*).

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
