## [Editor Report · eLife Assessment]

This **important** study provides **compelling** insights into the differential impact of intrinsic and synaptic conductances on circuit robustness using computational models of the pyloric network from the crustacean stomatogastric ganglion. The results demonstrate that model networks are more sensitive to perturbations in intrinsic conductances than in synaptic conductances, highlighting the critical role of intrinsic plasticity in stabilizing neuronal networks. These findings underscore the importance of intrinsic plasticity, a crucial yet often overlooked factor in neuronal dynamics. The generality of these conclusions should be tested across diverse networks and functions.

---

## [Referee Report · Reviewer #1 (Public review)]

The paper by Fournier et al. investigates the sensitivity of neural circuits to changes in intrinsic and synaptic conductances. The authors use models of the stomatogastric ganglion (STG) to compare how perturbations to intrinsic and synaptic parameters impact network robustness. Their main finding is that changes to intrinsic conductances tend to have a larger impact on network function than changes to synaptic conductances, suggesting that intrinsic parameters are more critical for maintaining circuit function.

The paper is well-written, and the results are compelling. The authors addressed most of the minor comments I had and improved the manuscript.

However, it remains unclear how general the results are and what the underlying mechanism is. Regarding generality, the authors changed the title and added a sentence in the discussion. At this point, they do not claim generality beyond the specific function they explore in the STG circuit. While this is acceptable, I still believe the paper would be much more insightful if it provided a more general statement and investigated the mechanism behind why, in their hands, synaptic parameters appear more resilient to changes than intrinsic parameters.

---

## [Author Response]

The following is the authors’ response to the original reviews

**Public Reviews:**

**Reviewer #1 (Public review):**
The paper by Fournier et al. investigates the sensitivity of neural circuits to changes in intrinsic and synaptic conductances. The authors use models of the stomatogastric ganglion (STG) to compare how perturbations to intrinsic and synaptic parameters impact network robustness. Their main finding is that changes to intrinsic conductances tend to have a larger impact on network function than changes to synaptic conductances, suggesting that intrinsic parameters are more critical for maintaining circuit function.The paper is well-written and the results are compelling, but I have several concerns that need to be addressed to strengthen the manuscript. Specifically, I have two main concerns:(1) It is not clear from the paper what the mechanism is that leads to the importance of intrinsic parameters over synaptic parameters.(2) It is not clear how general the result is, both within the framework of the STG network and its function, and across other functions and networks. This is crucial, as the title of the paper appears very general.I believe these two elements are missing in the current manuscript, and addressing them would significantly strengthen the conclusions. Without a clear understanding of the mechanism, it is difficult to determine whether the results are merely anecdotal or if they depend on specific details such as how the network is trained, the particular function being studied, or the circuit itself. Additionally, understanding how general the findings are is vital, especially since the authors claim in the title that "Circuit function is more robust to changes in synaptic than intrinsic conductances," which suggests a broad applicability.I do not wish to discourage the authors from their interesting result, but the more we understand the mechanism and the generality of the findings, the more insightful the result will be for the neuroscience community.Major comments(1) MechanismWhile the authors did a nice job of describing their results, they did not provide any mechanism for why synaptic parameters are more resilient to changes than intrinsic parameters. For example, from Figure 5, it seems that there is mainly a shift in the sensitivity curves. What is the source of this shift? Can something be changed in the network, the training, or the function to control it? This is just one possible way to investigate the mechanism, which is lacking in the paper.(2) Generality of the results within the framework of the STG circuit(a) The authors did show that their results extend to multiple networks with different parameters (the 100 networks). However, I am still concerned about the generality of the results with respect to the way the models were trained. Could it be that something in the training procedure makes the synaptic parameters more robust than intrinsic parameters? For example, the fact that duty cycle error is weighted as it is in the cost function (large beta) could potentially affect the parameters that are more important for yielding low error on the duty cycle.(b) Related to (a), I can think of a training scheme that could potentially improve the resilience of the network to perturbations in the intrinsic parameters rather than the synaptic parameters. For example, in machine learning, methods like dropout can be used to make the network find solutions that are robust to changes in parameters. Thus, in principle, the results could change if the training procedure for fitting the models were different, or by using a different optimization algorithm. It would be helpful to at least mention this limitation in the discussion.(3) Generality of the functionThe authors test their hypothesis based on the specific function of the STG. It would be valuable to see if their results generalize to other functions as well. For example, the authors could generate non-oscillatory activity in the STG circuit, or choose a different, artificial function, maybe with different duty cycles or network cycles. It could be that this is beyond the scope of this paper, but it would be very interesting to characterize which functions are more resilient to changes in synapses, rather than intrinsic parameters. In other words, the authors might consider testing their hypothesis on at least another 'function' and also discussing the generality of their results to other functions in the discussion.(4) Generality of the circuitThe authors have studied the STG for many years and are pioneers in their approach, demonstrating that there is redundancy even in this simple circuit. This approach is insightful, but it is important to show that similar conclusions also hold for more general network architectures, and if not, why. In other words, it is not clear if their claim generalizes to other network architectures, particularly larger networks. For example, one might expect that the number of parameters (synaptic vs intrinsic) might play a role in how resilient the function is with respect to changes in the two sets of parameters. In larger models, the number of synaptic parameters grows as the square of the number of neurons, while the number of intrinsic parameters increases only linearly with the number of neurons. Could that affect the authors' conclusions when we examine larger models?In addition, how do the authors' conclusions depend on the "complexity" of the non-linear equations governing the intrinsic parameters? Would the same conclusions hold if the intrinsic parameters only consisted of fewer intrinsic parameters or simplified ion channels? All of these are interesting questions that the authors should at least address in the discussion.

We thank Reviewer #1 for their valuable input. We agree with the reviewer that generality of the results may have been overstated. To address this we changed the title of the manuscript to make it more specific to rhythmic circuits and we included a sentence to this effect in the discussion.

(1) We were more interested in knowing which set of conductances is more robust in a population of models, rather than a mechanism. If such a mechanism exists it will be the subject of a different study.

(2) (a) It is impossible to explore the whole parameter space of these models. Our method to find circuits will leave subsets of circuits out of the study. Our sole goal in constructing the model database was that the activities were similar but the conductances were different. (b) Of course one could devise a cost function targeting circuits that are more or less robust to changes in one parameter. Whether those exist is a different matter. This is not what we intended to do.

(3) For this we would need a different circuit that produces non-oscillatory activity. A normal pyloric rhythm circuit always produces oscillatory activity unless it is “crashed"either by temperature or perturbations, but even in this case because we don’t have a proper “control” activity (circuits crash in different ways) we would not be able to utilize the same approach.

We think it is a valuable idea to perform a similar study in another small circuit with nonoscillatory (or rhythmic) activities.

(4) We did not explore the issue of how our results generalize to larger networks as it would be pure speculation. It could be potentially interesting to do a similar sensitivity analysis with a large network trained to perform a simple task. Our understanding is that many large trained networks are extremely sensitive to perturbations in synaptic weights, at the same time that the intrinsic properties of neurons in ANN are typically oversimplified and identical across units.

**Reviewer #2 (Public review):**
Summary:This manuscript presents an important exploration of how intrinsic and synaptic conductances affect the robustness of neural circuits. This is a well-deserved question, and overall, the manuscript is written well and has a logical progression.The focus on intrinsic plasticity as a potentially overlooked factor in network dynamics is valuable. However, while the stomatogastric ganglion (STG) serves as a well-characterized and valuable model for studying network dynamics, its simplified structure and specific dynamics limit the generalizability of these findings to more complex systems, such as mammalian cortical microcircuits.Strengths:Clean and simple model. Simulations are carefully carried out and parameter space is searched exhaustively.Weaknesses:(1) Scope and Generalizability:The study's emphasis on intrinsic conductance is timely, but with its minimalistic and unique dynamics, the STG model poses challenges when attempting to generalize findings to other neural systems. This raises questions regarding the applicability of the results to more complex circuits, especially those found in mammalian brains and those where the dynamics are not necessarily oscillating. This is even more so (as the authors mention) because synaptic conductances in this study are inhibitory, and changes to their synaptic conductances are limited (as the driving force for the current is relatively low).(2) Challenges in Comparison:A significant challenge in the study is the comparison method used to evaluate the robustness of intrinsic versus synaptic perturbations. Perturbations to intrinsic conductances often drastically affect individual neurons' dynamics, as seen in Figure 1, where such changes result in single spikes or even the absence of spikes instead of the expected bursting behavior. This affects the input to downstream neurons, leading to circuit breakdowns. For a fair comparison, it would be essential to constrain the intrinsic perturbations so that each neuron remains within a particular functional range (e.g., maintaining a set number of spikes). This could be done by setting minimal behavioral criteria for neurons and testing how different perturbation limits impact circuit function.(3) Comparative Metrics for Perturbation:Another notable issue lies in the evaluation metrics for intrinsic and synaptic perturbations. Synaptic perturbations are straightforward to quantify in terms of conductance, but intrinsic perturbations involve more complexity, as changes in maximal conductance result in variable, nonlinear effects depending on the gating states of ion channels. Furthermore, synaptic perturbations focus on individual conductances, while intrinsic perturbations involve multiple conductance changes simultaneously. To improve fairness in comparison, the authors could, for example, adjust the x-axis to reflect actual changes in conductance or scale the data post hoc based on the real impact of each perturbation on conductance. For example, in Figure 6, the scale of the panels of the intrinsic (e.g., g_na-bar) is x500 larger than the synaptic conductance (a row below), but the maximal conductance for sodium hits maybe for a brief moment during every spike and than most of the time it is close to null. Moreover, changing the sodium conductance over the range of 0-250 for such a nonlinear current is, in many ways, unthinkable, did you ever measure two neurons with such a difference in the sodium conductance? So, how can we tell that the ranges of the perturbations make a meaningful comparison?

We thank Reviewer #2 for their comments. We agree with both reviewers about scope and generalizability. We changed the title of the manuscript and included a sentence in the discussion to address this.

**Recommendations for the authors:**

**Reviewer #1 (Recommendations for the authors):**
(1) Line 63: Tau_b is tau in Fig 1B? What is the 'network period' tau_n? Both are defined in the methods, but it would be good to clarify here and also in the figure.

This was fixed. Tau_b is the bursting period and we indicated it in the figure. Network period means the period of the network activity. This was rewritten.

(2) Line 74: "maximal conductances g_i." What is i? I can imagine what you meant, but it would be good to clarify the notation.

There are multiple different currents. Letter ‘i' is an index over the different types. It now reads as follows,

"The activity of the network depends on the values of the maximal conductances g ¯ i, where i is an index corresponding to the different current types (Na,CaS,CaT,Kd,KCa,A,H,Leak IMI)"

(3) Line 78: "conductances are changed by a random amount." How much is the "random amount"? In percentages?

We fixed this sentence. This is how it reads now,

"The blue trace in Figure 1C corresponds to the activity of the same model when each of the intrinsic conductances is changed by a random amount within a range between 0 (completely removing the conductance) and twice its starting value, 2×gi, or equivalently, an increment of 100%."

Similarly, in Line 87: "by a similar percent." Can you provide Figures 1E-F in percentages? Are the percentages the same?

The phrase "by a similar percent.” Is misleading and unimportant. Thank you, we removed it.

(4) Line 113: Why did you add I_MI? Is it important for the results or for the conclusions?

I_MI was added because the current is known to be there and it is not more or less important for the results or conclusions than any other current.

(5) Line 117: "We used a genetic algorithm to generate a database." Confusing. I guess you meant that you used genetic algorithms to optimize the cost function.

Thank you for this comment. We fixed this sentence, see below.

“We used a genetic algorithm to optimize the cost function, and in this way generated a database of N = 100 models with different values of maximal conductances (Holland 88)."

(6) Line 136: "The models in the database were constrained to produce solutions whose features were similar to the experimental measurements." Why are there differences in the features? Is this an optimization issue? I thought you wanted to claim that there are degenerate solutions, that is, solutions where the parameters are different, but the output is identical. Please clarify.

The concept of degenerate solutions does not imply that the solutions are mathematically identical. In biology this means that they provide very similar functions, but do so with different underlying parameters (in this case, maximal conductances). The activity of the pyloric network is slightly different across animals, and it also changes over time within the same individual. Variation across models reflects individual variation in the biological circuit, and it is strength of our modeling approach. The function of the circuits are equally good because they produce biologically realistic patterns, although the details of the activity patterns show differences.

(7) Line 139: "distributed (p > 0.05)." What test did you use? N? Similarly, at Lines 218, 241, 239, etc. Please be more rigorous when reporting statistical tests.

Thank you. We now specify the test we utilized every time we report a p value.

(8) Line 143: "In this case, it is not possible to identify clusters, suggesting that there are no underlying relationships between the features in the model database." The 2D plot is misleading, as the features are in 11 dimensions. Claims should be about the 11D space, not projections onto 2D. In fact, I don't think you can rule out correlations between the features based on the 2D plots. For example, shouldn't there be correlations between the on and off phases and the burst durations?

Thank you. These sentences were confusing and were removed. We added the following sentence to the end of that paragraph.

"Because the feature vectors are similar, their t-SNE projections do not form groups or clusters."

(9) Related to this, I don't understand this sentence: "Even though the conductances are broadly distributed over many-fold ranges, the output of the circuits results in tight yet uncorrelated distributions.”

This sentence is confusing and was removed.

(10) Line 158: Repetition of Line 152: Figure 3 shows the currentscapes of each cell in two model networks.

We removed the second instance of the repeated sentences.

(11) Line 160: "yet the activity of the networks is similar." Well, they are similar, but not identical. I can also say that the current scapes are 'similar'. This should be better quantified and not left as a qualitative description.

While this is an interesting point it will not change the results and conclusions of the present study. The network models are different since the values of their maximal conductances are distributed over wide ranges.

(12) Line 218: midpoint parameter? Is that b - the sharpness? Please be consistent. Regarding the mechanism (see above) - any ideas what leads to this shift in the sensitivity curves between the two types of parameters?

Yes, we made a mistake. ‘b’ is the midpoint parameter. This was fixed in the text, thank you.

(13) Figure 6 illustrates why synaptic parameters are more robust, but it is not quantified. Why not provide a quantitative measure for this claim? For example, calculate the colored area within the white square for each pair, for each cell, and for each model. Show that these measures can predict improved robustness for one model over another and for synaptic vs. intrinsic parameters.

The ratio of areas of the colored and non-colored regions in the whole hyperboxes (for intrinsic and synaptic conductances) is the number reported in the y-axis of the sensitivity curves when we include all conductances (and not just a pair).

We computed the ratios of the colored/noncolored areas in all panels in figure 6 and now report these quantities as follows,

"We computed the proportions of areas of the white boxes that correspond to pyloric activity. These values for the intrinsic conductances panels are PD = 0.58, LP = 0.50, PY = 0.49, and the proportions for the synaptic conductances panels are PDPY = 0.62, P DLP = 0.87, and LPPD = 0.94. The occupied areas for synaptic conductances are larger than in the intrinsic conductances panels, consistent with our finding that the circuits’ activities are more robust to changes in synaptic conductances versus changes in intrinsic conductances."

"As before, we computed the proportion of areas of pyloric activity within the white boxes: PD = 0.61, LP = 0.55, PY = 0.52, and the proportions for the synaptic conductances panels are PDPY = 0.88, PDLP = 0.87, and LPP D = 0.83. These results provide an intuition of the complexities of GP . Not only are these regions hard-to-impossible to characterize in one circuit, but they are also different across circuits.”

(14) Does the sign of the synaptic weights affect the conclusions?

We did not explore this issue because all chemical synapses in this network are inhibitory.

(15) Line 492: typo: deltai.

We fixed this.

**Reviewer #2 (Recommendations for the authors):**
(1) Line 301 - you can also add Williams and Fletcher 2019 Neuron.

We added the reference. Thank you.

(2) Line 316 - this is a strange comment as these exact regions that were shown intrinsic plasticity (e.g., Losonczy, Attila, Judit K. Makara, and Jeffrey C. Magee. "Compartmentalized dendritic plasticity and input feature storage in neurons." Nature 452.7186 (2008): 436-441).

We did not understand this comment.

(3) I found only one citation for the work of Turrigiano, the most relevant of which is only mentioned in the Method section. This is odd, as her work directly relates how synaptic conductance perturbation results in changes in intrinsic conductance.

We included more references to the work of Turrigiano to provide more context.

"Desai, Niraj S., Lana C. Rutherford, and Gina G. Turrigiano. "Plasticity in the intrinsic excitability of cortical pyramidal neurons." Nature neuroscience 2, no. 6 (1999): 515-520.” "Desai, Niraj S., Sacha B. Nelson, and Gina G. Turrigiano. "Activity-dependent regulation of excitability in rat visual cortical neurons." Neurocomputing 26 (1999): 101-106.”

(4) Line 329 - The list of citations is very limited regarding studies of ext/int balance which started really way before 2009. Please give some of the credit to the classics.

We included the following additional references.

Van Vreeswijk, Carl, and Haim Sompolinsky. "Chaos in neuronal networks with balanced excitatory and inhibitory activity." Science 274, no. 5293 (1996): 1724-1726.

Rubin, Ran, L. F. Abbott, and Haim Sompolinsky. "Balanced excitation and inhibition are required for high-capacity, noise-robust neuronal selectivity." Proceedings of the National Academy of Sciences 114, no. 44 (2017): E9366-E9375.

Wang, Xiao-Jing. "Macroscopic gradients of synaptic excitation and inhibition in the neocortex." Nature reviews neuroscience 21, no. 3 (2020): 169-178.

Lo, Chung-Chuan, Cheng-Te Wang, and Xiao-Jing Wang. "Speed-accuracy tradeoff by a control signal with balanced excitation and inhibition." Journal of Neurophysiology 114, no. 1 (2015): 650-661.

(5) In Figure 1B, why does it say 'OFF' when the neuron is spiking?

The label indicates the interval of time elapsed between the first spike in the PD neuron (taken as a reference), and the last spike in the burst (PD off).

Summary of changes to figures:

Figure 1:

Fixed labels indicating bursting period and burst duration.

Figure 5:

Added labels in panels C and D specifying the symbol corresponding to the sigmoidal parameter.

Additional changes

We changed the title of the manuscript as follows:

"Rhythmic circuit function is more robust to changes in synaptic than intrinsic conductances." We included the following sentence at the end of the Discussion Section.

"We believe our results will hold for other rhythmic circuits and will be relevant for similar studies in other circuits with more complex functions.”

We realized we made a mistake with the units for maximal conductances. They were incorrectly expressed in nS (nano Siemens) in the figure labels, and correctly expressed in micro Siemens in the methods section. This was fixed and now conductances are expressed in micro Siemens consistently in the manuscript.